# Synergistic Interaction between Copper and Nitrogen-Uptake, Translocation, and Distribution in Rice Plant

**DOI:** 10.3390/plants11192612

**Published:** 2022-10-04

**Authors:** Xinlong Cui, Hua He, Shengwang Hu, Banfa Zhang, Hongmei Cai

**Affiliations:** 1Microelement Research Center, Huazhong Agricultural University, Wuhan 430070, China; 2College of Resources and Environment, Huazhong Agricultural University, Wuhan 430070, China

**Keywords:** Cu-N interplay, Cu transport, N transport, N assimilation, gene expression, plant nutrition

## Abstract

Interactions among nutrients have been widely recognized in plants and play important roles in crop growth and yield formation. However, the interplay of Cu and N in rice plants is not yet clear. In this study, rice plants were grown with different combinations of Cu and N supply. The effects of Cu-N interaction on the growth, yield production, Cu and N transport, and gene expression levels were analyzed. The results showed that the effect of N supply on rice growth and yield formation was more pronounced than that of Cu supply. The Cu supply significantly improved the uptake of N (by 9.52–30.64%), while the N supply significantly promoted the root-to-shoot translocation of Cu (by 27.28–38.45%) and distributed more Cu (1.85–19.16%) into the shoots and leaves. The results of qRT-PCR showed that +Cu significantly up-regulated the expression levels of both NO_3_^−^ and NH_4_^+^ transporter genes *OsNRTs* and *OsAMTs*, including *OsNRT1.1B*, *OsNRT2.1*, *OsNRT2.3a*, *OsNRT2.4*, *OsAMT1.2*, *OsAMT1.3*, and *OsAMT3.1*. Meanwhile, +N significantly up-regulated the expression levels of Cu transporter genes *OsHMA5* and *OsYSL16*. In addition, the supply of Cu up-regulated the expression levels of *OsGS1;2*, *OsGS2*, and *OsNADH-GOGAT* to 12.61-, 6.48-, and 6.05-fold, respectively. In conclusion, our study demonstrates a synergistic effect between Cu and N in rice plants. It is expected that our results would be helpful to optimize the application of N and Cu fertilizers in agriculture.

## 1. Introduction

Nitrogen (N) is a key essential macronutrient required for healthy plant growth and development, which is often a major limiting factor for crop yield production in agriculture [1,2]. N is the constituent of key cell molecules such as nucleic acids, amino acids, chlorophyll, ATP, and plant hormones. Therefore, N is important for many biological processes including photosynthesis, carbon metabolism, and amino acid and protein synthesis [2,3]. A lack of N may have a severe impact on plant growth and development, such as root branching, old leaf chlorosis, increased root-to-shoot ratio, reduced growth and photosynthesis, and fewer seed formation [4,5,6]. Nitrate (NO_3_^−^) and ammonium (NH_4_^+^) are two major forms of N that are absorbed by plant roots from the growth medium. Plants uptake and transport NO_3_^−^ through nitrate transporters (NRTs), while they uptake and transport NH_4_^+^ through ammonium transporters (AMTs) [3]. After absorbing by the plant root, NO_3_^−^ can be reduced into NH_4_^+^ through nitrate reductase (NR) and nitrite reductase (NiR). Then, the NH_4_^+^ is assimilated into amino acids for protein synthesis through the glutamine synthetase (GS)/glutamate synthase (GOGAT) cycle [1,7]. 

Copper (Cu) is one of the essential micronutrients for plant growth and development. Cu is also involved in multiple biological processes, such as photosynthesis, oxidative stress resistance, and ethylene perception [8,9,10]. Cu deficiency may cause specific symptoms in plants, including distortion or whitening of young leaves, apical meristem damage, and decreased seed setting and grain yield [8,9]. The Cu transporter (COPT), heavy-metal ATPase (HMA), yellow stripe-like protein (YSL), and Cu chaperones are reported to be the main transporters and proteins involved in the uptake, root-to-shoot translocation, distribution, and redistribution of Cu in plants [11,12,13,14,15,16]. 

To date, a large number of studies have revealed strong interactions existing between different nutrients that changing one or more nutrients in the growth medium may affect many other nutrients in plants. Recently, several studies have reported the synergistic interactions between N and other nutrients in plants. For example, the phosphate starvation response is strongly and actively controlled by N provision, and PHR1 (phosphate starvation response 1), PHO2 (phosphate overaccumulation 2), and NRT1.1 (nitrate transporter 1.1) are the major components that integrate N and phosphorus (P) signals in *Arabidopsis*, rice, and wheat [17]. In rice, the supply of potassium (K) increased N use efficiency by promoting the transformation of storage N to photosynthetic N in the leaves [18]. Molybdenum (Mo) plays a key role in both N uptake and assimilation in winter wheat. The contents of nitrite, ammonium, amino acids, and proteins were significantly increased by the application of Mo [19,20]. Moreover, a markedly synergistic effect of the co-application of N and boron (B) on N uptake, nitrogen use efficiency, and yield production has been shown in rapeseed under B-deficient conditions [21]. Zinc (Zn) and N can also synergistically act in rice plant. The co-supply of Zn and N significantly promoted the root-to-shoot translocation and preferential distribution of both N and Zn, thereafter improving the yield production in rice [22].

Rice is the main staple food with a total planting area of about 30.3 million hectares and feeding > 65% of the population in China [23,24]. Because of the growing population, limited arable land, climate vulnerability, environmental degradation, and low resource-use efficiency, much more effort is needed to improve the rice yield production and ensure food security in China than ever before [25,26,27]. In the last few decades, large amounts of N fertilizers have been applied to improve crop yield in China. However, the N use efficiency in crops is still relatively low, which only accounts for 30–50% in rice [27]. Therefore, increasing crop N use efficiency is still a critical issue in China. Nowadays, the concentration of Cu in agricultural soil has been elevated in various regions, as Cu is always added with pesticide and co-fertilized with other fertilizers, or added to soils by mining and smelting activities and industrial wastewater and solids [28,29,30,31,32]. Hence, in order to know how the co-existence of Cu and N affects the crop growth and yield production, we planted rice plants in both solutions and soils with different combinations of Cu and N supply. The growth phenotype, yield production, uptake, translocation, and distribution of Cu and N were studied to reveal the synergistic effect between Cu and N in rice plant. In addition, the expression levels of genes involved in Cu and N transport, and N reduction and assimilation were analyzed to reveal the underlying mechanism. Our study demonstrates a synergistic effect between Cu and N in rice plant. It is expected that our results would be helpful to optimize the application of N and Cu fertilizers in agriculture.

## 2. Results

### 2.1. Growth Phenotype of Rice Plant under the Combined Applications of Cu and N

To assess the effect of different combinations of Cu and N on rice growth, plants were grown in solutions with different combinations of Cu and N supply (Figure 1a, Appendix A). The root length gradually decreased with increasing levels of N under -Cu and +Cu conditions, but not under ++Cu conditions, and ++Cu significantly decreased the root length under all three N conditions (Figure 1b). Quite different results were shown in the plant height, shoot dry weight, and leaf SPAD (soil plant analysis development) value, which gradually increased with increasing levels of N under all three Cu conditions (Figure 1c,e; Appendix A). However, there was little effect of Cu on the plant height, shoot dry weight, and leaf SPAD value, except that the shoot dry weight significantly increased with increasing levels of Cu under -N conditions (Figure 1e). The root dry weight did not show significant differences under different combinations of Cu and N, except that it increased by +Cu treatment under -N conditions (Figure 1d).

### 2.2. Cu Improves the Uptake of N in Rice Plant

To investigate the effect of Cu on N transport in rice plant, we determined the N concentration in the root, basal node, and shoot samples, and calculated the N accumulation, root uptake capacity, and root-to-shoot translocation and distribution under different combinations of Cu and N supply. There was no significant effect of Cu on N concentration in the root, basal node, and shoot under -N, +N, or ++N conditions, except that a higher N concentration in the root after +Cu treatment under +N conditions, higher N concentration in the basal node after +Cu and ++Cu treatments under ++N conditions, and higher N concentration in the shoot after +Cu and ++Cu treatments under +N conditions were observed (Figure 2a). The N accumulation in both the basal node and shoot gradually increased with increasing levels of Cu under all three N conditions, while little effect of Cu on the N accumulation in the root was observed (Figure 2b). The supply of N significantly improved the root uptake capacity of N under various Cu conditions, while the supply of Cu clearly increased the root uptake capacity of N under +N and ++N conditions, but not under -N conditions (Figure 3a). Compared to -Cu treatments, the root uptake capacity of N increased by 30.31% and 10.56% after +Cu and ++Cu treatments, respectively, under +N conditions, and increased by 13.71% and 18.88% after +Cu and ++Cu treatments, respectively, under ++N conditions (Figure 3a). The supply of N significantly improved the plant uptake of N under various Cu conditions (Figure 3b). Simultaneously, the supply of Cu also significantly increased the plant uptake of N under all three N conditions (Figure 3b). Compared to -Cu treatment, the uptake of N in rice plant increased by 9.52% and 24.17% after +Cu and ++Cu treatments, respectively, under -N conditions, increased by 30.64% and 17.70% after +Cu and ++Cu treatments, respectively, under +N conditions, and increased by 10.05% and 18.86% after +Cu and ++Cu treatments, respectively, under ++N conditions (Figure 3b). The supply of N slightly improved the root-to-shoot translocation of N, while the supply of Cu did not affect the root-to-shoot translocation of N (Figure 3c). However, the distribution of N in different organs did not show any significant changes under different combinations of Cu and N supply (Figure 3d). These results suggest that the effect of Cu-N interaction on N uptake is more pronounced than the effect on the root-to-shoot translocation and distribution of N in rice plant. Our results demonstrate that the Cu supply can improve the uptake of N in rice plant, but not the root-to-shoot translocation of N or the distribution of N in each organ.

### 2.3. N Promotes the Translocation and Distribution of Cu in Rice Plant

There was little effect of N on the Cu concentration in the root, basal node, and shoot under -Cu, +Cu, or ++Cu conditions that only a lower Cu concentration in the root after +N treatment under ++Cu conditions, a lower Cu concentration in the basal node after +N and ++N treatments under +Cu and ++Cu conditions, but a higher Cu concentration in the shoot after ++N treatment under ++Cu conditions were observed (Figure 4a). The Cu accumulation in the root and basal node gradually increased with increasing N levels under -Cu conditions, and the Cu accumulation in the shoot gradually increased with increasing N levels under all three Cu conditions, while no significant differences were shown in the Cu accumulation of the root and basal node among different N treatments under either +Cu or ++Cu conditions (Figure 4b). The N supply did not affect the root uptake capacity of Cu under any Cu condition, except that a higher root uptake capacity of Cu was shown after ++N treatment under ++Cu conditions (Figure 5a). Similarly, the N supply did not affect the plant uptake of Cu under +Cu and ++Cu conditions, while an increasing plant uptake of Cu was shown with increasing N levels under -Cu conditions (Figure 5b). Interestingly, the N supply significantly increased the root-to-shoot translocation of Cu under +Cu and ++Cu conditions, but not under -Cu conditions (Figure 5c). Compared to -N treatment, the root-to-shoot translocation of Cu increased by 27.28% and 38.45% after +N and ++N treatments, respectively, under +Cu conditions, and increased by 29.41% and 34.27% after +N and ++N treatments, respectively, under ++Cu conditions (Figure 5c). The N supply also promoted the distribution of Cu in the rice shoot under all three Cu conditions (Figure 5d). Compared to -N treatment, the distribution of Cu in the rice shoot increased 3.87% and 1.85% after +N and ++N treatments, respectively, under -Cu conditions, increased 13.41% and 19.16% after +N and ++N treatments, respectively, under +Cu conditions, and increased 6.72% and 9.72% after +N and ++N treatments, respectively, under ++Cu conditions (Figure 5d). Our results demonstrate that the N supply is beneficial for distributing more Cu to the shoot, which may promote the growth of rice plant. 

### 2.4. Cu Up-Regulated the Expression Levels of Genes Involved in N Transport, Reduction, and Assimilation

To understand the molecular mechanism of the effect of Cu supply on N uptake, reduction, and assimilation, we analyzed the expression levels of key genes involved in N transport, reduction, and assimilation in both root and shoot samples under different combinations of Cu and N supply. The results showed that the expression levels of most of the nitrate and ammonium transport genes as well as nitrate reduction and ammonium assimilation genes were up-regulated by Cu addition, except that the expression levels of two low-affinity nitrate transport genes, *OsNRT1.1A* and *OsNRT1.1B*, were down-regulated by Cu addition in the root under -N conditions (Figure 6a–c). The effect of Cu on the expression of nitrate and ammonium transport genes was more pronounced in the root than in the shoot. For example, the expression levels of *OsNRT2.1* and *OsNRT2.3a* increased to 12.41-fold and 5.16-fold, respectively, after +Cu treatment in the root under -N conditions, while the expression levels of *OsNRT1.1B* and *OsNRT2.4* increased to 13.85-fold and 6.89-fold, respectively, after +Cu treatment in the root under +N conditions (Figure 6a). The expression level of *OsAMT1.2* increased to 8.39-fold after +Cu treatment in the root under -N conditions, and the expression levels of *OsAMT1.2*, *OsAMT1.3*, and *OsAMT3.1* increased to 6.25-fold, 5.11-fold, and 5.89-fold, respectively, after +Cu treatment in the root under +N conditions (Figure 6b). The effect of Cu on the expression of ammonium assimilation genes was more pronounced than the nitrate reduction genes. For example, the expression level of *OsGS1;2* increased to 12.62-fold after +Cu treatment in the root under -N conditions, the expression level of *OsNADH-GOGAT* increased to 6.05-fold after +Cu treatment in the root under +N conditions, and the expression level of *GS2* increased to 6.48-fold after +Cu treatment in the shoot under -N conditions (Figure 6c). 

### 2.5. N Up-Regulated the Expression Levels of Genes Involved in Cu Transport

The results of the expression analysis of genes involved in the Cu transport also showed that the expression levels of most of the Cu transport genes were up-regulated by N addition. The effect of N on the expression of Cu transport genes was more pronounced in the root than in the shoot. For example, the expression levels of *OsCOPT1* and *OsHMA5* increased to 3.88-fold and 22.98-fold, respectively, after +N treatment in the root under -Cu conditions, and the expression levels of *OsCOPT7, OsHMA5*, and *OsYSL16* increased to 8.59-fold, 3.83-fold, and 3.73-fold, respectively, after +N treatment in the root under +Cu conditions (Figure 6d). Only the expression level of *OsHMA5* increased to 4.16-fold after +N treatment in the shoot under -Cu conditions (Figure 6d).

### 2.6. Yield Evaluation of Rice Plant under the Combined Applications of Cu and N

To investigate the effect of different combinations of Cu and N on rice yield formation, plants were grown in soils with different combinations of Cu and N supply (Figure 7a, Appendix A). The N supply significantly increased the straw dry weight, grain yield, and number of effective panicles under both -Cu and +Cu conditions, while it only increased the plant height, panicle length, fertility, and seed number per panicle under -Cu conditions, but it did not affect the 1000-seed weight under either -Cu or +Cu conditions (Figure 7b,c; Appendix A). However, there was less impact of Cu on the plant growth and yield formation. The Cu supply could increase the panicle length, fertility, and seed number per panicle, but significantly decreased the 1000-seed weight and number of effective panicles under all three N conditions (Appendix A). The Cu supply only increased the plant height and grain yield under -N conditions, and no significant changes were observed in straw dry weight after Cu supply under any treatment of N (Figure 7c). 

### 2.7. N Affects The Concentration, Accumulation, and Distribution of Cu in Rice Plant at Mature Stage

Unexpectedly, the Cu supply did not affect the N concentration, accumulation, and distribution in rice plants at the mature stage (data not shown). However, the N supply significantly affected the Cu concentration, accumulation, and distribution in rice plants at this growth stage (Figure 8). The Cu concentrations in all organs, including new leaf, old leaf, node, internode, spike stalk, and grain, gradually increased with the increasing levels of N under both -Cu and +Cu conditions (Figure 8a). Compared to -N treatment, the Cu accumulation in rice plant increased by 152.21% and 166.35% after +N and ++N treatments, respectively, under -Cu conditions, and increased by 37.72% and 138.03% after +N and ++N treatments, respectively, under +Cu conditions (Figure 8b). The N supply significantly increased the Cu distribution in leaves but decreased the Cu distribution in rice grains under both -Cu and +Cu conditions (Figure 8c). Compared to -N treatment, the Cu distribution in leaves increased 6.94% and 4.03% after +N and ++N treatments, respectively, under -Cu conditions, and increased 12.02% and 10.37% after +N and ++N treatments, respectively, under +Cu conditions (Figure 8c). The Cu distribution in grains decreased 10.73% and 7.83% after +N and ++N treatments, respectively, under -Cu conditions, and decreased 14.06% and 15.07% after +N and ++N treatments, respectively, under +Cu conditions (Figure 8c). Our results suggest that N supply improves both the concentration and accumulation of Cu in rice plant, and also promotes the distribution of Cu in the shoot at the mature stage, which may be beneficial for the yield production.

## 3. Discussion

Interactions among nutrients have been widely recognized in plants. Previous studies have shown that synergistic effects between N and several other nutrients, such as P, K, Mo, B, and Zn, existed in crops [17,18,19,20,21,22]. Here, we studied the interplay between Cu and N in rice plant. Our results also showed a synergistic effect between Cu and N existing in rice plant. The supply of Cu improved the uptake of N, while the supply of N promoted the root-to-shoot translocation of Cu and distributed more Cu into shoots and leaves, which is beneficial for the plant growth and yield production in rice (Figure 1, Figure 3, Figure 5 and Figure 7). However, no significant impact of N supply on Cu concentration in rice plant was observed in our study, while Cai et al. reported that the foliar Cu concentration increased with the addition of N in forbs and grasses [33]. Similarly, no significant impact of Cu supply on the N concentration in rice plant was observed in our study, while Zhou et al. reported that the application of N increased the Cu content in the roots of R. *communis* [34]. The interactive effect between Cu and N may be various in plants with different application levels of Cu or N. The negative impact of N supply (280 kg ha^−1^) on Cu uptake was observed in N. *nivea* grown in Cu-contaminated soils [35]. Furthermore, high levels of Cu inhibited the N uptake and upward translocation in Chinese cabbage, and influenced N metabolism in rice plant [36,37]. 

*NRTs* and *AMTs* are the two major gene families encoding the membrane transporters contributed for NO_3_^−^ and NH_4_^+^ uptake and transportation in plants. In our study, the expression levels of *OsNRT2.1* and *OsNRT2.3a* significantly increased (>5-fold) after +Cu treatment in the root under -N conditions, while the expression levels of *OsNRT1.1B* and *OsNRT2.4* significantly increased (>5-fold) after +Cu treatment in the root under +N conditions (Figure 6a). *OsNRT2.1* and *OsNRT2.3a* are two high-affinity nitrate transporter genes mainly expressed in the rice root and involved in NO_3_^−^ uptake and long-distance transport, respectively, under low-N conditions [38,39]. *OsNRT1.1B* is a low-affinity nitrate transporter gene mainly expressed in rice roots and contributed to NO_3_^−^ uptake under N-sufficient conditions [40], while OsNRT2.4 behaves as a dual-affinity nitrate transporter and plays important roles in NO_3_^−^ uptake under both N-deficient and -sufficient conditions [41]. Moreover, the expression level of *OsAMT1.2* increased to 8.39-fold after +Cu treatment in the root under -N conditions, and the expression levels of *OsAMT1.2*, *OsAMT1.3*, and *OsAMT3.1* increased over 5-fold after +Cu treatment in the root under +N conditions (Figure 6b). *OsAMT1;2* is a root-specific and ammonium-inducible expressed gene, whereas *OsAMT1;3* is also expressed specifically in roots but repressed by N, both of them being involved in NH_4_^+^ uptake in rice plant [42,43,44,45]. Our results clearly showed that the Cu supply significantly up-regulated the expression levels of *OsNRTs* and *OsAMTs* in rice plant, and the impact was more pronounced in the root than in the shoot (Figure 6a,b). Therefore, the higher expression levels of these N transporter genes in the root may contribute to the improved N uptake in rice plant after Cu addition. 

In addition to N transport genes, our results also showed that the Cu supply significantly up-regulated the expression levels of genes involved in nitrate reduction and ammonium assimilation, and the impact of Cu on the expression of ammonium assimilation genes was more pronounced than the nitrate reduction genes (Figure 6c). The expression level of *OsGS1;2* and *OsNADH-GOGAT* significantly increased after +Cu treatment in the root under -N and +N conditions, respectively, and the expression level of *GS2* significantly increased after +Cu treatment in the shoot under -N conditions (Figure 6c). *OsGS1;2* and *OsNADH-GOGAT* are two root-expressed genes encoding the cytosolic glutamine synthetase and glutamate synthase, respectively, which are important for the primary ammonium assimilation in rice root [46,47,48], while *OsGS2* is a leaf-expressed gene encoding the chloroplastic glutamine synthetase and is a main factor involved in primary ammonium assimilation in rice leaves [46,49]. Our results suggested that the Cu supply might also promote nitrate reduction and ammonium assimilation in rice plant through up-regulating the expression levels of *OsGS1;2* and *OsNADH-GOGAT* in the root as well as the expression level of *OsGS2* in rice leaf. 

OsHMA5 is a plasma-membrane-localized heavy-metal ATPase, which is highly expressed in the root pericycle cells and xylem region of diffuse vascular bundles in nodes, and functions in loading Cu to the xylem of the roots, and translocating Cu to aboveground organs [14]. OsYSL16 is a Cu-nicotianamine transporter, which is highly expressed in the phloem of nodes and vascular tissues of leaves, and functions in delivering Cu to the developing young tissues in rice plant [15,50]. In our study, the expression levels of *OsHMA5* increased to 22.98-fold and 4.16-fold after +N treatment in the root and shoot, respectively, under -Cu conditions, and the expression levels of *OsHMA5* and *OsYSL16* increased to 8.59-fold and 3.73-fold, respectively, after +N treatment in the root under +Cu conditions (Figure 6d). These results suggested that the N supply promoted the root-to-shoot translocation of Cu and distribution of more Cu into the shoots and leaves mainly through up-regulating the expression levels of *OsHMA5* and *OsYSL16*. Although a significantly higher expression level of *OsCOPT7* (8.59-fold) was also observed in the rice root after +N treatment under +Cu conditions (Figure 6d), the basal expression level of *OsCOPT7* in the rice root is relatively low as reported by Yuan et al. [51], which means that the function of OsCOPT7 in Cu uptake is relatively weak, and that is why the uptake of Cu was not affected by N supply in our study. 

In conclusion, our study showed a synergistic effect between Cu and N in rice plant. The Cu supply improved the uptake of N through up-regulating the expression levels of both NO_3_^−^ and NH_4_^+^ transporter genes *OsNRTs* and *OsAMTs*, while the N supply promoted the root-to-shoot translocation of Cu and distributed more Cu into the shoots and leaves through up-regulating the expression levels of Cu transporter genes *OsHMA5* and *OsYSL16*. In addition, the Cu supply significantly up-regulated the expression levels of primary ammonium assimilation genes *OsGS1;2*, *OsGS2*, and *OsNADH-GOGAT*, which may enhance the N metabolic level in rice plant. Although our results may be helpful to optimize the application of N and Cu fertilizers in agriculture, the field experiment is still needed in the future to determine appropriated amounts of N and Cu fertilizers for achieving the maximum N use efficiency and yield production.

## 4. Materials and Methods

### 4.1. Hydroponic Culture

The rice cultivar Guangliangyou 35, one of the most widely planted hybrid indica rice cultivars in Hubei Province, China, was used in this study. The experiment was conducted in Huazhong Agricultural University, Wuhan, China (30°46′ N, 114°36′ E) in March 2021. For hydroponic culture, seeds were soaked in water at 30 °C in the dark for three days and then transferred to a net floating on a 0.5 mM CaCl_2_ solution. After a week, the seedlings were transferred to a Yoshida solution containing 1.44 mM NH_4_NO_3_, 0.3 mM NaH_2_PO_4_, 0.5 mM K_2_SO_4_, 1.0 mM CaCl_2_, 1.6 mM MgSO_4_, 0.17 mM Na_2_SiO_3_, 50 μM Fe-EDTA, 0.06 μM (NH_4_)_6_Mo_7_O_24_, 15 μM H_3_BO_3_, 8 μM MnCl_2_, 0.15 μM CuSO_4_, 0.15 μM ZnSO_4_, 29 μM FeCl_3_, and 40.5 μM citric acid (pH 5.6) [52]. The different treatments of CuSO_4_ (-Cu, 0 μM; +Cu, 0.15 μM; ++Cu, 1.5 μM) and NH_4_NO_3_ (-N, 0.144 mM; +N, 1.44 mM; ++N, 7.20 mM) were conducted. Seedlings were grown in a greenhouse at 25 °C to 30 °C under natural light. Solutions were renewed every three days. Three biological replicates with two plants for each replicate were conducted. Three weeks later, the roots were washed three times with 0.5 mM CaCl_2_. The root, basal node, and shoot samples were harvested for the determination of Cu and N, and the root and shoot samples were harvested for the gene expression analysis by qRT-PCR as described below. 

### 4.2. Pot Experiment

The experiment was conducted in Huazhong Agricultural University, Wuhan, China (30°46′ N, 114°36′ E) from July to October in 2021. For pot experiment, seeds were soaked in water at 30 °C in the dark for three days and then transferred to a net floating on a 0.5 mM solution of CaCl_2_. After a week, seedlings were transferred to pots containing 10 kg of dry soil, which was fertilized with 0.15 g P_2_O_5_ kg^−1^ soil and 0.2 g K_2_O kg^−1^ soil. The different treatments of CuSO_4_ (-Cu, 0 mg Cu kg^−1^ soil; +Cu, 50 mg Cu kg^−1^ soil) and urea (-N, 0 g N kg^−1^ soil; +N, 0.2 g N kg^−1^ soil; ++N, 0.4 g N kg^−1^ soil) were conducted. Seedlings were grown at 25 °C to 35 °C under natural light until maturation. Three biological replicates were conducted. Four rice plants were planted for each biological replicate. At the mature stage, the grain yield was evaluated, and different organs, including the new leaf, old leaf, node, internode, spike stalk, and grain, were harvested for the determination of Cu and N as described below.

### 4.3. N Determination

The harvested samples were dried in an oven at 70 °C for 5 days. After recording the dry weight, the samples were ground into powder. Amounts of 0.05 g from each sample were subjected to digestion with 3 mL of 98% H_2_SO_4_ on a heater at up to 280 °C. In addition, 2-3 drops of H_2_O_2_ were added several times until the solution became transparent. The concentration of total N in the digested solution was determined using a flow injection analyzer (FIAstar 5000, FOSSTECATOR, Stockholm, Sweden). The N accumulation was calculated as N concentration × dry weight; the N uptake was calculated as the N accumulation of whole plant; the uptake capacity of N was calculated as the N accumulation of whole plant/root dry weight; the root-to-shoot translocation of N was calculated as the N accumulation in shoot/the N accumulation of whole plant; the N distribution in each organ was calculated as the N accumulation in each organ/the N accumulation of whole plant.

### 4.4. Cu Determination

The harvested samples were dried in an oven at 70 °C for 5 days. After recording the dry weight, the samples were ground into powder. Amounts of 0.15 g from each sample were subjected to digestion with 3 mL of 11 N HNO_3_ on a heater at up to 130 °C until the solution became transparent. The concentration of Cu in the digested solution was determined using an atomic absorption spectrophotometer (WFX-ID, RUILI, Beijing, China). The Cu accumulation was calculated as Cu concentration × dry weight; the Cu uptake was calculated as the Cu accumulation of whole plant; the uptake capacity of Cu was calculated as the Cu accumulation of whole plant/root dry weight; the root-to-shoot translocation of Cu was calculated as the Cu accumulation in shoot/the Cu accumulation of whole plant; the Cu distribution in each organ was calculated as the Cu accumulation in each organ/the Cu accumulation of whole plant.

### 4.5. RNA Extraction and qRT-PCR Analysis

Total RNA was extracted from root and shoot samples using a RNeasy Plant Mini Kit (Yeasen, Shanghai, China) and converted to cDNA using the Rever Tra Ace qPCR RT Master Mix with a gDNA remover (Yeasen, Shanghai, China) following the manufacturer’s protocol. The cDNA was amplified using the SYBR Green Real-Time PCR Master Mix Kit (KAPA, Boston, USA) following the manufacturer’s protocol. The reaction was added with 2.0 μL of cDNA, 0.2 μL of forward primer, 0.2 μL of reverse primer, 5.0 μL of SYBR Green, and 2.6 μL of ddH_2_O. The quantitative real-time PCR was performed using a Quant Studio TM 6 Flex System (Applied Biosystems, Foster City, CA, USA) with specific gene primers (Appendix A). The reaction was performed as 95 °C 10 s, 60 °C 20 s, and 72 °C 20 s, for 40 cycles. The relative gene expression levels were normalized using an internal standard (*OsUbiquitin*) and calculated using the 2^−ΔΔCt^ method with CFX Manager Software (Bio-Rad, CA, USA). The fold changes of N-related genes were calculated as the expression levels of each gene under +Cu conditions/the expression levels of each gene under -Cu conditions; the fold changes of Cu-related genes were calculated as the expression levels of each gene under +N conditions/the expression levels of each gene under -N conditions.

### 4.6. Statistical Analysis 

For statistical analyses, the software of SPSS 19.0 (Statistical Package for the Social Sciences, version of 19.0, Chicago, IL, USA) was used. Analysis of variance (ANOVA) was employed to identify significant differences for each parameter among the treatments of different combined applications of Cu and N at the 5% probability level (*p* < 0.05). Data are presented as means ± SD with three independent replicates. Significant differences were determined by Tukey’s multiple range test (*p* < 0.05). 

## Figures and Tables

**Figure 1 plants-11-02612-f001:**
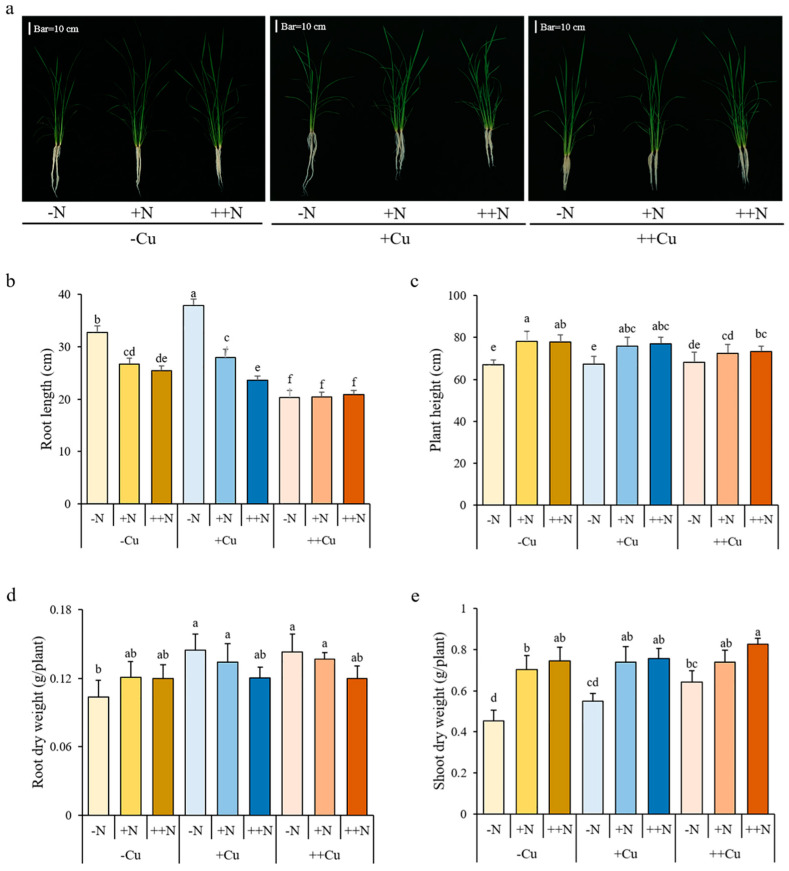
Growth phenotype (**a**), root length (**b**), plant height (**c**), and root (**d**) and shoot (**e**) dry weight of rice plant grown in solutions with different combinations of Cu (-Cu, 0 μM; +Cu, 0.15 μM; ++Cu, 1.5 μM) and N (-N, 0.288 mM; +N, 2.88 mM; ++N, 14.4 mM) supply for three weeks. Data are means ± SD of three biological replicates. Different letters indicate significant difference at *p* < 0.05 by Tukey’s test.

**Figure 2 plants-11-02612-f002:**
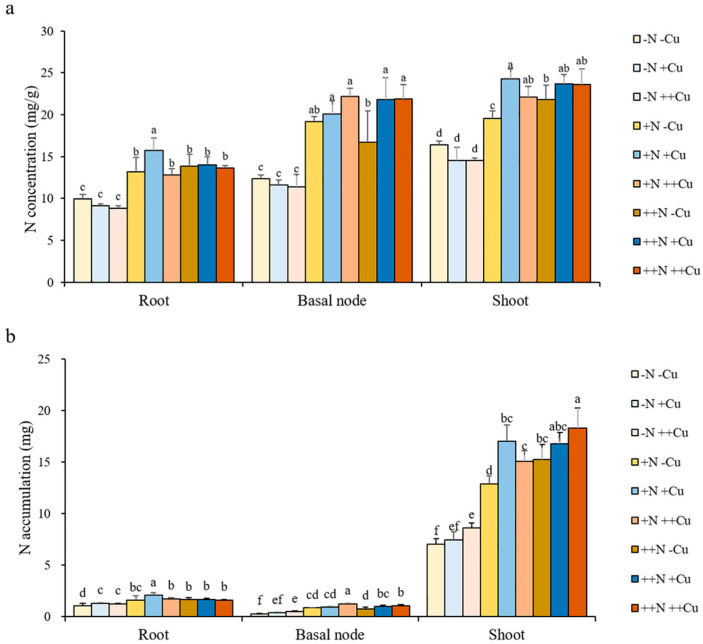
N concentration (**a**) and accumulation (**b**) in the root, basal node, and shoot of rice plant grown in solutions with different combinations of Cu (-Cu, 0 μM; +Cu, 0.15 μM; ++Cu, 1.5 μM) and N (-N, 0.288 mM; +N, 2.88 mM; ++N, 14.4 mM) supply for three weeks. The N accumulation was calculated as N concentration × dry weight. Data are means ± SD of three biological replicates. Different letters indicate significant difference at *p* < 0.05 by Tukey’s test.

**Figure 3 plants-11-02612-f003:**
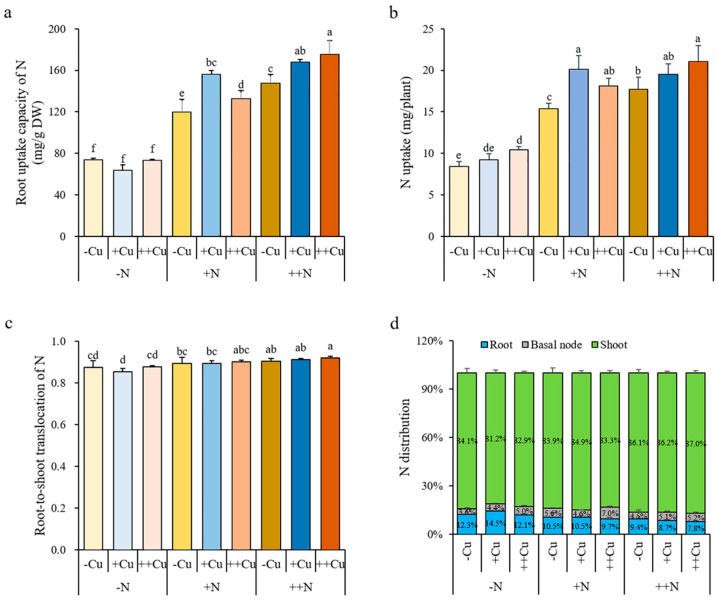
Root uptake capacity of N (**a**), N uptake in whole plant (**b**), root-to-shoot translocation of N (**c**), and N distribution (**d**) of rice plant grown in solutions with different combinations of Cu (-Cu, 0 μM; +Cu, 0.15 μM; ++Cu, 1.5 μM) and N (-N, 0.288 mM; +N, 2.88 mM; ++N, 14.4 mM) supply for three weeks. The N uptake was calculated as the N accumulation of whole plant; the uptake capacity of N was calculated as the N accumulation of whole plant/root dry weight; the root-to-shoot translocation of N was calculated as the N accumulation in shoot/the N accumulation of whole plant; the N distribution in each organ was calculated as the N accumulation in each organ/the N accumulation of whole plant. Data are means ± SD of three biological replicates. Different letters indicate significant difference at *p* < 0.05 by Tukey’s test.

**Figure 4 plants-11-02612-f004:**
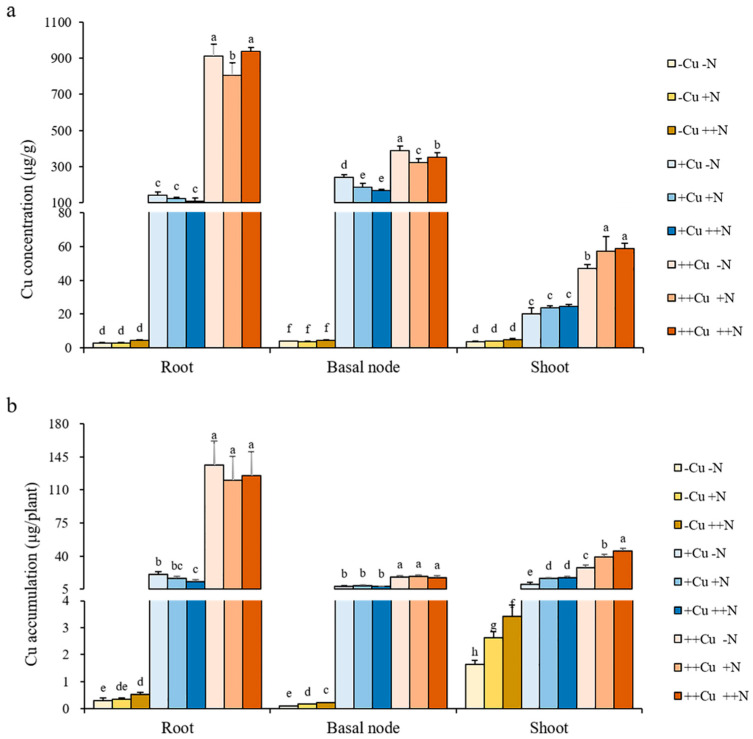
Cu concentration (**a**) and accumulation (**b**) in the root, basal node, and shoot of rice plant grown in solutions with different combinations of Cu (-Cu, 0 μM; +Cu, 0.15 μM; ++Cu, 1.5 μM) and N (-N, 0.288 mM; +N, 2.88 mM; ++N, 14.4 mM) supply for three weeks. The Cu accumulation was calculated as Cu concentration × dry weight. Data are means ± SD of three biological replicates. Different letters indicate significant difference at *p* < 0.05 by Tukey’s test.

**Figure 5 plants-11-02612-f005:**
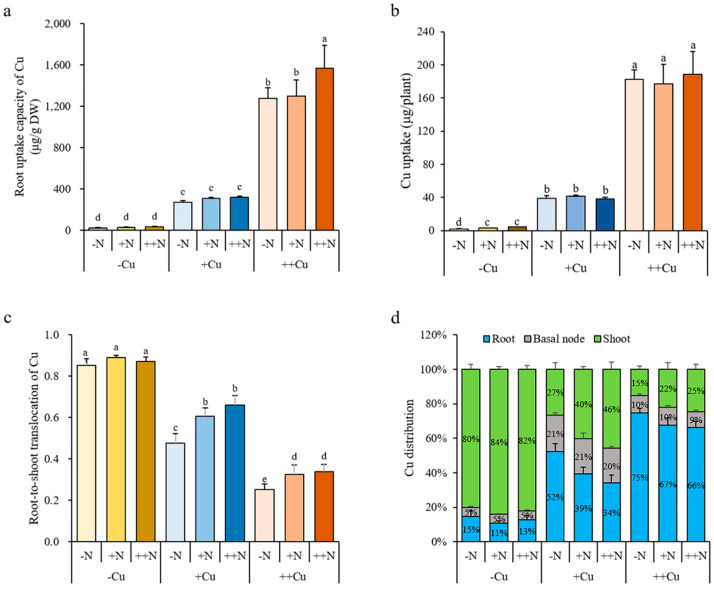
Root uptake capacity of Cu (**a**), Cu uptake in whole plant (**b**), root-to-shoot translocation of Cu (**c**), and Cu distribution (**d**) of rice plant grown in solutions with different combinations of Cu (-Cu, 0 μM; +Cu, 0.15 μM; ++Cu, 1.5 μM) and N (-N, 0.288 mM; +N, 2.88 mM; ++N, 14.4 mM) supply for three weeks. The Cu uptake was calculated as the Cu accumulation of whole plant; the uptake capacity of Cu was calculated as the Cu accumulation of whole plant/root dry weight; the root-to-shoot translocation of Cu was calculated as the Cu accumulation in shoot/the Cu accumulation of whole plant; the Cu distribution in each organ was calculated as the Cu accumulation in each organ/the Cu accumulation of whole plant. Data are means ± SD of three biological replicates. Different letters indicate significant difference at *p* < 0.05 by Tukey’s test.

**Figure 6 plants-11-02612-f006:**
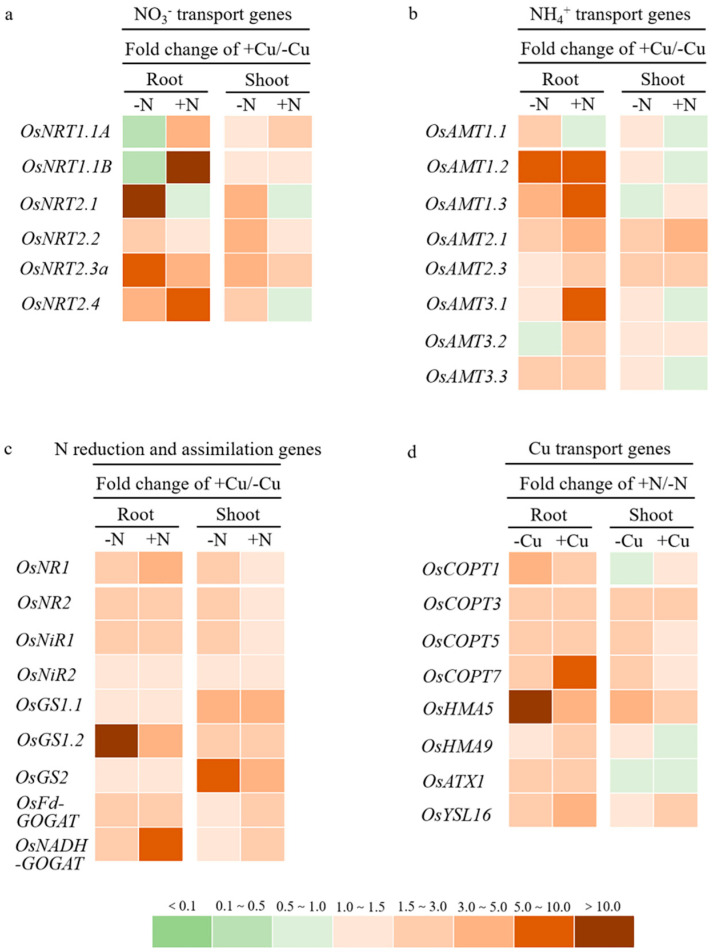
Expression levels of genes involved in NO_3_^−^ transport (**a**), NH_4_^+^ transport (**b**), NO_3_^−^ reduction and NH_4_^+^ assimilation (**c**), and Cu transport (**d**) in the root and shoots of rice plant grown in solutions with different combinations of Cu (-Cu, 0 μM; +Cu, 0.15 μM) and N (-N, 0.288 mM; +N, 2.88 mM) supply for three weeks. Data are fold changes of expression levels of N-related genes under +Cu compared to -Cu conditions, or Cu-related genes under +N compared to -N conditions.

**Figure 7 plants-11-02612-f007:**
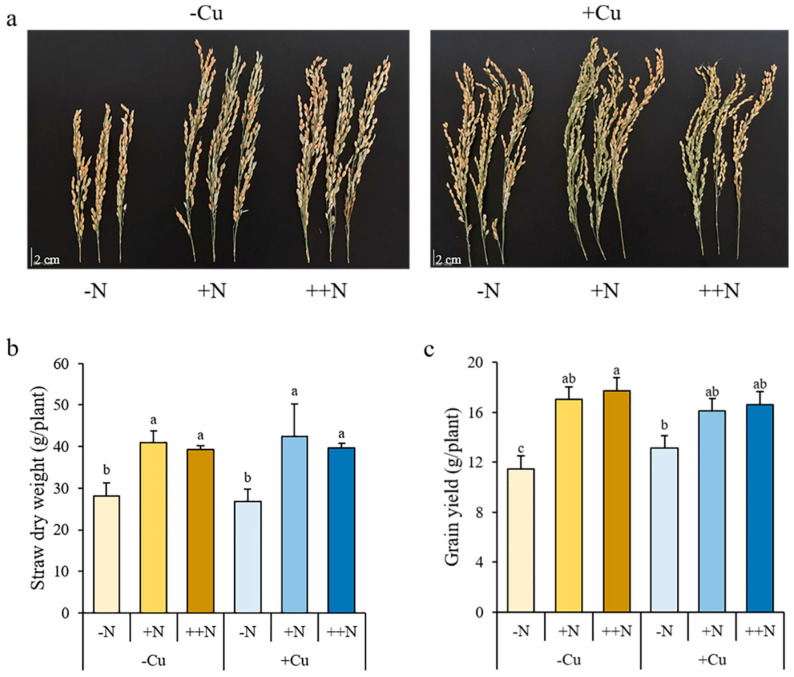
The panicles (**a**), straw dry weight (**b**), and grain yield (**c**) of rice plant grown in soil with different combinations of Cu (-Cu, 0 mg Cu kg^−1^ soil; +Cu, 50 mg Cu kg^−1^ soil) and N (-N, 0 g N kg^−1^ soil; +N, 0.2 g N kg^−1^ soil; ++N, 0.4 g N kg^−1^ soil) supply at mature stage. Data are means ± SD of three biological replicates. Different letters indicate significant difference at *p* < 0.05 by Tukey’s test.

**Figure 8 plants-11-02612-f008:**
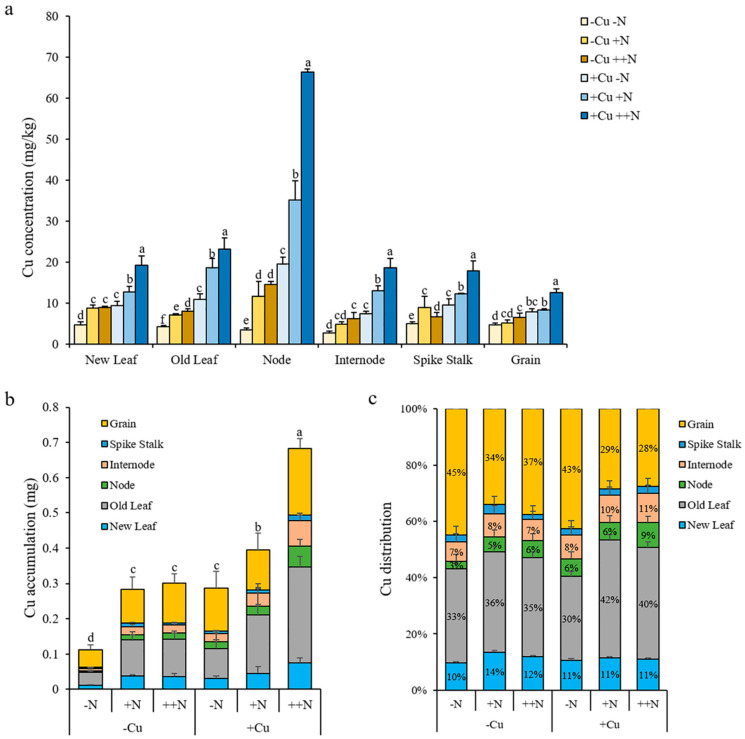
The concentration (**a**), accumulation (**b**), and distribution (**c**) of Cu in rice plant grown in soil with different combinations of Cu (-Cu, 0 mg Cu kg^−1^ soil; +Cu, 50 mg Cu kg^−1^ soil) and N (-N, 0 g N kg^−1^ soil; +N, 0.2 g N kg^−1^ soil; ++N, 0.4 g N kg^−1^ soil) supply at mature stage. The Cu accumulation was calculated as Cu concentration × dry weight; the Cu distribution in each organ was calculated as the Cu accumulation in each organ/the Cu accumulation of whole plant. Data are means ± SD of three biological replicates. Different letters indicate significant difference at *p* < 0.05 by Tukey’s test.

## Data Availability

Not applicable.

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
