# Peer review of "Synergistic Interaction between Copper and Nitrogen-Uptake, Translocation, and Distribution in Rice Plant"

_plants, 2022, doi:10.3390/plants11192612_

Round 1
Reviewer 1 Report
The present manuscript revealed the synergistic effect between Cu and N in rice plant. Rice plants were grown in both solutions and soils with different combinations of Cu and N supply, and the growth phenotype and yield production, the uptake, translocation, and distribution of Cu and N, as well as the expression levels of genes involved in Cu and N transport, N reduction and assimilation were analyzed. Results showed the effect of N supply on rice growth and yield formation was more pronounced than the Cu. The Cu supply significantly improved the uptake of N, while the N supply significantly promoted the root-to-shoot translocation of Cu and distributed more Cu into shoots and leaves. Results of qRT-PCR showed that +Cu significantly up-regulated the expression levels of both NO3- and NH4+ transporter genes OsNRTs and OsAMTs, as well as the primary NH4+ assimilation genes OsGS1;2, OsGS2, and OsNADH-GOGAT, while +N significantly up-regulated the expression levels of Cu transporter genes OsHMA5 and OsYSL16.
In general, the work is well-performed and the results are important and useful for improving rice growth and yield. However, I suggest the following minor revisions;
- Mention the main objective of the present study clearly in both of the abstract and end of introduction section.
- Explain how you performed the analyses and parameters measurements clearly in the methods section.
- Why you selected those genes analyzed by real-time qPCR?
- Figure 3 needs more explanation and revealing the hypothesis behind it.
- Mention the main future recommended work at the end of the conclusion.
- Revise and make sure that the references format adhere to the journal style.
- An overall English language revisions is recommended.
Author Response
The present manuscript revealed the synergistic effect between Cu and N in rice plant. Rice plants were grown in both solutions and soils with different combinations of Cu and N supply, and the growth phenotype and yield production, the uptake, translocation, and distribution of Cu and N, as well as the expression levels of genes involved in Cu and N transport, N reduction and assimilation were analyzed. Results showed the effect of N supply on rice growth and yield formation was more pronounced than the Cu. The Cu supply significantly improved the uptake of N, while the N supply significantly promoted the root-to-shoot translocation of Cu and distributed more Cu into shoots and leaves. Results of qRT-PCR showed that +Cu significantly up-regulated the expression levels of both NO3- and NH4+ transporter genes OsNRTs and OsAMTs, as well as the primary NH4+ assimilation genes OsGS1;2, OsGS2, and OsNADH-GOGAT, while +N significantly up-regulated the expression levels of Cu transporter genes OsHMA5 and OsYSL16.
In general, the work is well-performed and the results are important and useful for improving rice growth and yield. However, I suggest the following minor revisions;
- Mention the main objective of the present study clearly in both of the abstract and end of introduction section.
Response: We have mentioned the main objective of our study in both of the abstract and end of introduction section.
- Explain how you performed the analyses and parameters measurements clearly in the methods section.
Response: We have revised the section of “Materials and methods” to explain them more clearly.
- Why you selected those genes analyzed by real-time qPCR?
Response: As the supply of Cu affected the N uptake, we would like to know whether Cu affected the N uptake genes. NRTs and AMTs are the two major gene families encoding the membrane transporters contributed for NO3- and NH4+ uptake in plants. Therefore, we analyzed the NRT genes which are function known and involved in the NO3- uptake, as well as the genes in AMT family involved in the NH4+ uptake. More N uptaken by plant may affect the NO3- reduction and NH4+ assimilation. The selected NR, NiR, GS, and GOGAT genes are all reported to be mainly involved in these processes in rice. Similarly, the N supply affected the translocation and distribution of Cu, then we analyzed the COPTs, HMAs, YSL16, and ATX1, which have been reported to be mainly involved in these processes in rice. Actually, we have analyzed all the 7 COPT genes. However, the analysis of COPT2/4/6 was failed, as their expression levels are quite low in rice plant.
- Figure 3 needs more explanation and revealing the hypothesis behind it.
Response: We have revised it.
- Mention the main future recommended work at the end of the conclusion.
Response: We have mentioned the future recommended work at the end of the conclusion.
- Revise and make sure that the references format adhere to the journal style.
Response: We have revised the format of references.
- An overall English language revisions is recommended.
Response: We have revised this manuscript by a native English speaker.
Reviewer 2 Report
The manuscript plants-1939827, entitled “Synergistic interaction between copper and nitrogen - uptake, translocation and distribution in rice plant” reported and discussed the results of a laboratory experiment where the synergic effect of nitrogen and copper plant nutrition was assessed on rice by a hydroponic and a pot experiment. In particular, two of copper and thee of nitrogen different concentration were tested. The effects on plant were assessed by measuring plant phenotyping and production, Cu and N concentration, genes expression.
In general, the manuscript and the experimental activity carried out seem to be of good quality following a strict scientific logic and according to widely used methods which have made it possible to obtain reliable results followed by a good presentation and discussion.
In my opinion, only minor changes are needed before publication, regarding typos, image quality and M&M clarifications.
Abstract: should be revised in order to be more simple to read.
Keyword: try to do not repeat words already cited in the title.
Introduction: it is of good quality and gives an appropriate description of the state of the art, as well as, present the relevance of this study.
Results: are clear written and simple to follow.
Materials and methods: Some aspect should be deepened in particular regarding the statistical analysis.
Discussion: is well articulated considering each aspect of the experimental activity.

Author Response
The manuscript plants-1939827, entitled “Synergistic interaction between copper and nitrogen - uptake, translocation and distribution in rice plant” reported and discussed the results of a laboratory experiment where the synergic effect of nitrogen and copper plant nutrition was assessed on rice by a hydroponic and a pot experiment. In particular, two of copper and thee of nitrogen different concentration were tested. The effects on plant were assessed by measuring plant phenotyping and production, Cu and N concentration, genes expression.
In general, the manuscript and the experimental activity carried out seem to be of good quality following a strict scientific logic and according to widely used methods which have made it possible to obtain reliable results followed by a good presentation and discussion.
In my opinion, only minor changes are needed before publication, regarding typos, image quality and M&M clarifications.
Abstract: should be revised in order to be more simple to read.
Response: We have revised it.
Keyword: try to do not repeat words already cited in the title.
Response: We have revised it.
Introduction: it is of good quality and gives an appropriate description of the state of the art, as well as, present the relevance of this study.
Response: Thank you very much for your comments.
Results: are clear written and simple to follow.
Response: Thank you very much for your comments.
Materials and methods: Some aspect should be deepened in particular regarding the statistical analysis.
Response: We have revised it.
Discussion: is well articulated considering each aspect of the experimental activity.
Response: Thank you very much for your comments.